# Agreement and Associations between Countermovement Jump, 5-Time Sit-To-Stand, Lower-Limb Muscle Power Equations, and Physical Performance Tests in Community-Dwelling Older Adults

**DOI:** 10.3390/jcm13123380

**Published:** 2024-06-08

**Authors:** Hélio José Coelho-Júnior, Samuel da Silva Aguiar, Ivan de Oliveira Gonçalves, Alejandro Álvarez-Bustos, Leocadio Rodríguez-Mañas, Marco Carlos Uchida, Emanuele Marzetti

**Affiliations:** 1Department of Geriatrics, Orthopedics and Rheumatology, Center for Geriatric Medicine (Ce.M.I.), Università Cattolica del Sacro Cuore, Largo F. Vito 1, 00168 Rome, Italy; 2University Centre UDF, Faculty of Physical Education, 704/904 Seps Eq 702/902, Brasília 70390-045, Brazil; ssaguiar0@gmail.com; 3Graduate Program in Physical Education, Federal University of Mato Grosso, Av. Fernando Corrêa da Costa 2367 Bairro Boa Esperança, Cuiabá 78060-900, Brazil; 4Department of Health, Piaget University, Av. Mogi das Cruzes 1001, Suzano 08673-010, Brazil; ivanogedfisica@gmail.com; 5Centro de Investigación Biomédica en Red Sobre Fragilidad y Envejecimiento Saludable (CIBERFES), Instituto de Salud Carlos III, Av. Monforte de Lemos 3-5, 28029 Madrid, Spain; a.alvarezbu@gmail.com (A.Á.-B.); leocadio.rodriguez@salud.madrid.org (L.R.-M.); 6Applied Kinesiology Laboratory, School of Physical Education, University of Campinas, Av. Érico Veríssimo 701, Campinas 13083-851, Brazil; uchida@unicamp.br; 7Fondazione Policlinico Universitario “Agostino Gemelli” IRCCS, Largo A. Gemelli 8, 00168 Rome, Italy

**Keywords:** physical performance, sarcopenia, frailty, muscle strength, balance

## Abstract

**Objectives**: The present study examined the agreement and associations of the 5-time sit-to-stand (5STS) test, the countermovement jump test, and lower-limb muscle power equations with a set of physical performance tests in older adults. **Methods:** Five hundred and thirty-four community-dwelling older adults were recruited for the study. Lower-limb muscle power measures included 5STS, the countermovement jump test, and muscle power equations. Isometric handgrip strength, timed “up-and-go!”, the 6 min walking test, one-leg stand, and walking speed at usual and fast paces were used to assess physical performance. Pearson’s correlations and Bland–Altman analyses were conducted to examine associations among muscle power measures. Linear and multiple regressions were run to explore associations of 5STS, the countermovement jump test, and muscle power equations with physical performance tests. **Results:** Weak correlations were observed among lower-limb muscle power measures. Bland–Altman results indicated important differences among the countermovement jump test, 5STS, and muscle power equations. Results of multiple linear regressions indicated that 5STS, the countermovement jump test, and muscle power equations were significantly associated with measures of muscle strength and mobility. However, only 5STS was significantly associated with balance. **Conclusions:** Our results indicate that the performance on the countermovement jump test and 5STS is weakly correlated with lower-limb muscle power equations. The only exception was the correlation found between the countermovement jump test and relative muscle power, highlighting the importance of accounting for body mass in muscle power evaluations. Muscle power measures were similarly associated with performance on handgrip strength, timed “up-and-go!”, and the 6 min walking test. The exclusive association of 5STS with balance suggests that a reassessment of 5STS muscle power equations may be warranted.

## 1. Introduction

Muscle power is the capacity to generate strength rapidly [1,2]. During aging, muscle power declines earlier and to a greater extent than other physical performance parameters (e.g., muscle strength) [3,4,5]. In older adults, muscle power is a strong and independent predictor of mobility disability, poor functional status, falls, hospitalization, and death [4,6,7,8,9]. Comparative studies have found that this physical capacity is a better indicator of musculoskeletal health [3], physical performance [4], mobility [5], and functional independence [6] than muscle strength. As such, muscle power should be regularly monitored in older adults to identify those at higher risk of negative health-related events.

The measurement of muscle power entails the use of complex equipment, which impacts clinical applicability. However, physical performance tests exist that allow muscle power to be estimated. The countermovement jump (CMJ) test, for example, consists of jumping as high as possible after a brief stretching of hips and knees. This test is a valid measure of neuromuscular status [10] and has been widely used to estimate lower-limb muscle power in several populations, including older adults [11,12,13,14].

More recently, equations to estimate lower-limb muscle power measures based on the performance of the 5-time sit-to-stand (5STS) test and anthropometric parameters were validated [6,15]. These equations have been associated with many health parameters and may predict negative events [6,8,15,16]. Nonetheless, the extent to which equation-based estimations of muscle power and CMJ can be considered comparable measures of muscle power is yet to be determined. Indeed, the possibility that these assessment tools might reflect different facets of muscle power is supported by a recent study that observed significant differences between muscle power estimated based on 5STS equations and a linear encoder [17].

Lower-limb muscle power is acknowledged as a proxy of overall physical performance [1] given its associations with important physical function measures, including muscle strength [18], balance [19,20], and mobility [5,21]. This highlights the importance of actively monitoring lower-limb muscle power using valid, feasible, and comparable assessment methods. Noticeably, some studies did not find significant associations between 5STS muscle power equations and other physical capacities (i.e., mobility and muscle strength) [17,22]. These results suggest that more studies are necessary before muscle power estimated from 5STS is included in a routine clinical examination of older adults.

To this end, the present study examined the agreement between CMJ and 5STS performance and lower-limb muscle power equations in a relatively large sample of community-dwelling older adults. We also compared the associations of these muscle power measures with a set of physical performance tests.

## 2. Materials and Methods

This was an observational cross-sectional study that examined the correlations between CMJ and 5STS performance and lower-limb muscle power equations in community-dwelling older adults. Furthermore, we examined the associations of these muscle power measures with a set of physical performance tests. The study was approved by the Research Ethics Committee of the University of Mogi das Cruzes (UMC, São Paulo, Brazil). All procedures were conducted in compliance with the Declaration of Helsinki and Resolution 196/96 of the National Health Council. The article was prepared according to the Strengthening the Reporting of Observational Studies in Epidemiology criteria [23].

### 2.1. Participant Recruitment

The recruitment of participants was conducted by convenience from January 2015 to January 2018 at a senior center located in the metropolitan area of São Paulo, Poá, Brazil. The study was promoted using posters strategically placed in public spaces, including parks, city halls, government buildings, bus stops, and train stations. Local radio broadcasts and newspapers were used to further advertise the study. Individuals were also extended invitations to partake through direct communication. To be deemed eligible, candidates had to meet the following criteria: being 60 years of age or older, demonstrating self-sufficiency, and possessing the capability to successfully fulfill all measurements outlined in the protocol. There were no additional criteria for selection. Prior to enrollment, each participant provided written informed consent. The study sample included 534 participants.

### 2.2. Anthropometry and Disease Conditions

Body height and mass were measured through a stadiometer and an analog medical scale, respectively. The body mass index (BMI) was then calculated as the ratio between body mass (kg) and the square of height (m^2^). Information on medical conditions was gathered through self-reporting and a careful examination of the community senior center’s medical charts.

### 2.3. Countermovement Jump

The CMJ was performed using a jump platform (Jump System Pro, Cefise, Brazil). Before testing, participants maintained an upright standing position, with feet parallel to each other at shoulder width and hands placed on the hips. On the signal “Go!”, participants flexed their knees at approximately 90° and jumped as high as possible [11,12,13,14]. No specific recommendations were provided to indicate hip position during the CMJ, given that this approach is not commonly used in the literature [11,12,13,14] and that older adults might require different hip adequations to stand up [24] according to their physical limitations.

### 2.4. 5-Time Sit-To-Stand Test and Muscle Power Measures

The 5STS test involved rising from a chair five times as quickly as possible with arms folded across the chest. Timing began when participants raised their buttocks off the chair and stopped when they were seated at the end of the fifth stand. Time performance was quantified using a stopwatch (Vollo Sports, São Paulo, Brazil). The test reliability in the present study was 0.8 or more (κ = 0.97).

Absolute, relative, and allometric muscle power values were estimated according to the equations proposed by Alcazar et al. [7]:
(1)$$\text{Absolute muscle power}\;(W)=\frac{Body\;mass\;\left(kg\right)\;\times \;0.9\;\times \;g\;\times \;\left[height\;\left(m\right)\;\times \;0.5\;-\;chair\;height\;\left(m\right)\right]}{no.\;of\;STS\;repetitions\;\times \;0.5}$$
(2)$$\text{Relative muscle power}\;(W/\text{kg})=\frac{Absolute\;muscle\;power}{Body\;mass}$$
(3)$$\text{Allometric muscle power}\;(W/m^{2})=\frac{Absolute\;muscle\;power}{Height\;squared}$$

### 2.5. Physical Performance Tests

The procedures to conduct the physical performance tests used in the present study are thoroughly described elsewhere [25,26]. A researcher detailed the operational procedures and demonstrated the tests. Then, a familiarization trial was performed to ensure that participants had fully understood each test. Except for the one-leg stand test and the CMJ, tests were performed twice with the best result being used for analysis.

The isometric handgrip strength (IHG) test was performed with participants sitting comfortably on a chair with their shoulders in a neutral position. The arm being assessed (dominant arm) had the elbow flexed at 90° near the torso, and the hand neutral with thumb up. A maximal contraction was performed over four seconds using a Jamar handheld hydraulic dynamometer (Fred Sammons Inc., Brookfield, IL, USA). The timed “up-and-go!” (TUG) test involved standing up from a chair without the help of the arms, walking, as fast as possible a distance of three meters around a marker placed on the floor, coming back to the same position, and sitting back on the chair. The test began when the researcher gave a “Go!” command. The stopwatch was activated when participants stood up from the chair and was stopped when they were seated again. The 6 min walk test (6MWT) was conducted indoors on a 30 m track. After remaining seated for 15 min, participants were asked to walk on the track as fast as possible for six minutes. The distance walked was recorded in meters and used for the analysis. Static balance was assessed via the one-leg stand test. The test was performed with participants standing in a unipodal stance with the dominant lower limb, the contralateral knee remaining flexed at 90°, the arms crossed in front of the chest, and the head straight, for up to 30 s. A stopwatch was activated when participants raised their contralateral foot off the floor and was stopped when the foot touched the floor again. Walking speed was measured over three meters. Participants were required to walk five meters at their usual and fastest possible cadence (without running). Before testing, participants stood with both feet on the starting line. Timing began when a foot reached the 1 m line and stopped when a foot reached the 4 m line. The 1 m intervals at the beginning and at the end of the course were used to avoid early acceleration and deceleration, respectively.

### 2.6. Statistical Analysis

The Kolmogorov–Smirnov test was used to verify the normal distribution of data. Data for continuous variables are presented as mean (± standard deviation) or median (interquartile range) for normally and non-normally distributed values, respectively. Absolute numbers (percentages) were used to summarize categorical variables. Associations between CMJ and 5STS performance and muscle power equations were tested using Pearson’s correlation (crude data) and Bland–Altman analyses (log10 transformed data). Linear and multiple regression analyses were conducted to test the associations of CMJ and 5STS muscle power measures with physical performance tests. Multiple regression was adjusted according to several potential confounders. In Model 1, the analysis was adjusted for age, sex, and BMI. In Model 2, the analysis was adjusted according to Model 1 plus the presence of comorbidities that might directly or indirectly influence physical function (e.g., hypertension, diabetes mellitus, joint pain, osteoarthritis, cardiovascular disease). Significance was set at 5% (*p* < 0.05) for all tests. All analyses were performed using SPSS software (version 23.0, SPSS Inc., Chicago, IL, USA).

## 3. Results

### 3.1. Participants Characteristics

The main characteristics of the 534 participants are shown in Table 1. Mean age was 67.6 ± 6.3 years, and 88.4% were women. Average BMI values (27.7 ± 5.3 kg/m^2^) indicated that most participants were overweight or obese. The 5STS performance ranged from 4.2 s to 20.7 s, with mean values of 10.8 s, not indicative of probable sarcopenia [27]. Mean muscle power values according to 5STS equations were higher than the cutoff values for the Brazilian population [28].

### 3.2. Pearson’s Correlation

Pearson’s tests were conducted to examine the correlations between CMJ, 5STS, and 5STS muscle power measures. CMJ was significantly and negatively correlated with 5STS (r = −0.15, *p* = 0.0001) and positively with absolute (r = 0.13, *p* = 0.01) and relative muscle power (r = 0.16, *p* = 0.0001). No significant correlations were observed with allometric muscle power (r = 0.07, *p* = 0.08). All significant correlations were classified as weak according to Pearson’s correlation coefficients.

### 3.3. Bland–Altman Results

The Bland–Altman results are shown in Figure 1. The highest biases (average discrepancies between methods) were observed for differences between CMJ and absolute muscle power (−1.23), and CMJ and allometric muscle power (0.78). The lowest biases were observed for comparisons between CMJ and relative muscle power (0.61), and CMJ and 5STS performance (−0.02). Data extrapolated both upper and lower limits of agreement, with special attention to the bottom left space. In addition, a trend for larger differences between CMJ and the other methods is noted according to improvements in test performance.

### 3.4. Linear and Multiple Regression

The results of linear and multiple regression analyses for the associations between muscle power measures and physical performance tests are shown in Table 2. In the crude model, CMJ, 5STS, absolute, relative, and allometric muscle power were significantly associated with IHG, TUG, and 6MWT performances. CMJ, 5STS, and relative muscle power were also significantly associated with balance performance. No significant associations were observed between muscle power measures and walking speed at either normal or fast pace. Results remained significant for the associations with IHG, TUG, and 6MWT when analyses were adjusted according to covariates. However, in the fully adjusted model, only 5STS was significantly associated with balance.

## 4. Discussion

Findings of the present study indicate that CMJ is only weakly correlated with 5STS performance and lower-limb muscle power estimated by equations. The largest differences were observed for absolute and allometric muscle power, whereas small differences were found for relative muscle power and 5STS. The visual analysis indicated that data extrapolated both upper and lower limits of agreement, with a trend for larger differences between methods according to increases in test performance. When the associations of lower-limb muscle power measures and physical performance were tested, similar results were obtained, except for the static balance test, which was only significantly associated with 5STS.

Previous studies have reported significant differences between CMJ and multijoint dynamic tests. Haff et al. [29] and Young et al. [30] found non-significant weak-to-moderate correlations between CMJ height and peak force during maximal strength tests. Nuzzo et al. [31] confirmed these findings by describing that CMJ performance (height) was not correlated with absolute dynamic measures of muscle strength, whereas significant moderate correlations were detected when muscle strength performance was adjusted for body mass. These findings are in line with our results regarding the associations between relative, but not absolute, muscle power and CMJ performance, and may be explained in the light of the second Newton’s law, given that acceleration—body movement—is the product of force output (muscle strength) produced to move the body mass vertically outside the platform or to stand up [31].

To create an explosive movement capable of producing maximal displacement of the body mass outside the platform, CMJ involves the activation of a stretch-shortening mechanism with elongation of the leg-extensor muscle tendon apparatus before a full extension of hips, knees, and ankles, to propel the center of mass vertically, causing a flight phase [32]. The stretch-shortening strategy serves to accumulate elastic energy during the eccentric contraction to produce an immediate subsequent explosive and powerful concentric contraction [33,34,35]. The 5STS test does not involve a stretch-shortening cycle and initiates with the test person seated and relaxed [36,37]. To stand up, the trunk is flexed moving the center of mass forward and outside the base of support [24,36,37], which requires full activation of knee and hip extensors and some degree of plantar flexion to produce strength and power to move the body forward (a peculiarity of this test) and vertically in a fast manner [24,36,37]. Unlike CMJ, hamstrings are co-activated at the end of the 5STS move [24,38], probably to stop the movement at the end of concentric actions, avoid a flight phase, and prevent power generation across the entire movement.

Hence, the weak correlations and amount of data extrapolated observed when CMJ was compared to 5STS-derived absolute muscle power are possibly explained by the fact that, although both measures involve muscle power during knee and hip extension, only CMJ includes an elastic component and a full explosive movement across concentric actions. The adjustment of absolute muscle power by body mass relativizes the measures, given that both are expected to represent an evaluation of the amount of force required to move a determined amount of mass.

We observed significant associations between lower-limb muscle power measures and physical performance. These findings are in line with several previous studies [18,36,37,38,39] and support the view of experts in the field [1,2,39] and internationally recognized associations [40,41] that this physical capacity should be actively monitored in older adults. Bean et al. [5] examined data from the InCHIANTI study and found significant associations between leg power and global physical performance. Similar findings were obtained by Kuo et al. [42] after examining data from the National Health and Nutrition Examination Survey. These results were further reinforced by Suzuki et al. [4] who identified significant associations between plantar flexion/dorsiflexion peak power and physical performance tests.

The associations between lower-limb muscle power and mobility tests (e.g., TUG and 6MWT) may be attributed to the necessity of participants to move fast during time-dependent tasks and promptly generate strength to maintain balance in moments of perturbations [5]. On the other hand, the observed correlations between upper-limb muscle strength and muscle power are indicative of the role muscle strength plays in determining muscle power. Indeed, IHG has been acknowledged as a proxy of overall muscle strength [43] and has been used as a tool for identifying neuromuscular diseases (e.g., sarcopenia) [27].

We found that only 5STS was significantly associated with static balance. The CMJ possibly involves a small balance component, given that the test person’s base of support remains stable throughout the test. It is possible that more balance is required if knee and hip flexions occur near 90° degrees, requiring the activity of more synergistic muscles, but this scenario is unlikely to occur in older adults [44]. The 5STS, instead, includes the motion of the center of mass forward and upward, necessitating prompt and appropriate balance functioning [20,45].

Unexpectedly, no correlation was found between 5STS-based muscle power values and static balance. One would anticipate muscle power to be related to balance owing to the fact that the ability of muscles to contract rapidly is utilized to regain stability in the face of balance challenges [20]. The 5STS-based muscle power equations have faced criticism due to their assumption that only a portion of the body mass accelerates during the concentric phase of movement, whereas the production of mechanical power in similar conditions should account for alterations in the kinetic and potential energy of the entire body [46]. Authors have also highlighted conceptual problems with the definition of the concentric phase [46]. Notably, both aspects of the 5STS-based muscle power equations criticized by Fabrica and Biancardi [47] are associated with the concentric phase of the movement, in which more balance is likely to be required [45]. Therefore, a reassessment of the 5STS muscle power equations may be warranted to take into consideration that the full body is moving during the concentric phase and/or review the components of the formula after conducting a deeper biomechanical analysis.

Alternatively, the one-leg stand test might not represent the best assessment tool to evaluate balance in older adults, especially in those with fear of falling and/or a history of falls, overweight or obesity, visual and proprioception problems, and dizziness. Indeed, this test involves remaining in a unipodal stance for up to 30 s, which might be difficult for older adults. The use of more complex analysis, such as baropodometry, could provide more detailed information on balance, including load distribution on feet during challenges [47]. Moreover, the one-leg stand test only provides a measure of static balance, and different results might be obtained if balance was evaluated using dynamic tests.

The findings of the present study have important clinical implications. The weak correlations found between muscle power assessment methods indicate that existing tools for estimating this physical function probably capture distinct facets. This scenario highlights the need for careful consideration when comparing investigations that assessed muscle power using different methods and emphasizes the need for additional studies examining the validity of available instruments and to explore potential adjustments that could enhance their accuracy. The presence of small differences between relative muscle power and CMJ emphasizes the importance of accounting for body mass in muscle power evaluations and offers valuable information for future research.

The present study has limitations that should be acknowledged. First, our sample included apparently healthy community-dwelling older adults with preserved physical function, and extrapolations of results to physically impaired individuals should be made with caution. Second, comparisons between CMJ and 5STS performance and muscle power equations were possible in our sample but are not feasible in individuals with mobility limitations. In fact, the visual inspection of Bland–Altman plots suggests that differences between the methods increase according to improvements in test performance, which likely occurs due to a ceiling effect in CMJ performance in older adults. Therefore, studies using other standard muscle power measurement systems (e.g., linear encoder) are needed. Third, body composition, particularly muscle mass, was not assessed in the present study. Fourth, participants were not evaluated for sarcopenia or frailty. Fifth, walking speed tests were conducted over a short distance (3 m), which might impede the identification of differences in normal- and fast-pace performance, mainly due to an insufficient production of power. Finally, the cross-sectional design of the study does not allow for inferring temporal or cause–effect relationships between the variables considered.

## 5. Conclusions

The main findings of the present study indicate that CMJ and 5STS performance and lower-limb muscle power equations are only weakly correlated, which suggests that existing tools for estimating this physical function probably capture distinct facets of muscle power. The presence of small differences between relative muscle power and CMJ emphasizes the importance of accounting for body mass in muscle power evaluations and offers valuable information for future research. When the associations between muscle power measures and physical performance were tested, muscle power demonstrated significant associations with IHG, TUG, and 6MWT, whereas only 5STS performance was associated with balance. These results might indicate that 5STS-based equations to estimate muscle power need to be refined. Hence, further investigations employing a more comprehensive balance analysis are warranted to ascertain the reliability and clinical usefulness of 5STS-based muscle power equations.

## Figures and Tables

**Figure 1 jcm-13-03380-f001:**
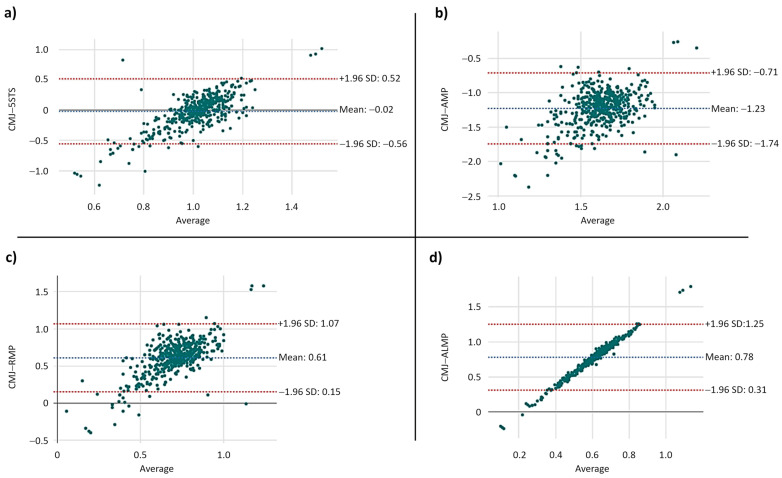
Bland–Altman plots between countermovement jump (CMJ) and (**a**) 5-time sit-to-stand (5STS) performance; (**b**) absolute muscle power (AMP); (**c**) relative muscle power (RMP); (**d**) allometric muscle power (ALMP).

**Table 1 jcm-13-03380-t001:** Characteristics of study participants (*n* = 534).

Age, years	67.6 ± 6.3
Women (%)	472 (88.4)
Body mass, kg	75.3, 35.3–99.7
Height, m	1.65, 1.37–1.93
Body mass index, kg/m^2^	27.7 ± 5.3
5-time sit-to-stand, s	10.8 ± 2.1
Absolute muscle power, W	177.3 ± 61.6
Relative muscle power, W/kg	2.5 ± 0.7
Allometric muscle power, W/m^2^	70.2 ± 26.5
Countermovement jump, cm	11.5 ± 8.1
Handgrip strength, kg	24.2 ± 6.8
One-leg stand, s	15.4 ± 11.9
Timed “up-and-go!”, s	7.5 ± 2.0
6 min walking test, m	568.4 ± 152.3
Walking speed at normal pace, m/s	1.0 ± 2.7
Walking speed at fast pace, m/s	1.0 ± 1.0
Hypertension (%)	351 (65.7)
Cardiovascular disease (%)	48 (9.0)
Diabetes mellitus (%)	92 (17.2)
Osteoarthritis (%)	229 (43.0)

Data for continuous variables are shown as mean ± standard deviation or median, interquartile range. Categorical data are presented as absolute values (percentage).

**Table 2 jcm-13-03380-t002:** Associations of muscle power with physical performance measures.

	Univariate β (95% CI)	*p*	Adjusted β ^1^ (95% CI)	*p*	Adjusted β ^2^ (95% CI)	*p*
Handgrip strength						
Countermovement jump	0.01 (0.08, 0.29)	0.000	0.12 (0.06, 0.18)	0.000	0.19 (0.08, 0.29)	0.000
5-time sit-to-stand	−0.46 (−0.74, −0.19)	0.000	−0.36 (−0.59, −0.12)	0.003	−0.35 (−0.64, −0.05)	0.019
Absolute muscle power	0.03 (0.02, 0.04)	0.000	0.02 (0.01, 0.03)	0.000	0.02 (0.01, 0.03)	0.000
Relative muscle power	2.58 (1.83, 3.34)	0.000	1.52 (0.85, 2.19)	0.000	1.25 (0.46, 2.03)	0.002
Allometric muscle power	0.06 (0.03, 0.08)	0.000	0.05 (0.02, 0.07)	0.000	0.04 (0.01, 0.06)	0.003
One-leg stand						
Countermovement jump	0.26 (0.13, 0.38)	0.000	0.18 (0.05, 0.31)	0.005	0.07 (−0.13, 0.28)	0.491
5-time sit-to-stand	−1.67 (−2.18, −1.15)	0.000	−1.54 (−2.04, −1.04)	0.000	−1.29 (−1.92, −0.66)	0.000
Absolute muscle power	0.01 (−0.00, 0.03)	0.146	0.02 (0.00, 0.04)	0.007	0.01 (−0.01, 0.03)	0.297
Relative muscle power	2.78 (1.31, 4.25)	0.000	2.11 (0.65, 3.57)	0.005	1.02 (−0.61, 2.65)	0.220
Allometric muscle power	0.02 (−0.02, 0.06)	0.397	0.06 (0.01, 0.12)	0.008	0.03 (−0.02, 0.08)	0.266
Timed “up-and-go!”						
Countermovement jump	−0.07 (−0.09, −0.05)	0.000	−0.05 (−0.08, −0.03)	0.000	−0.04 (−0.06, −0.02)	0.000
5-time sit-to-stand	0.33 (0.25, 0.40)	0.000	0.31 (0.23, 0.38)	0.000	0.24 (0.19, 0.29)	0.000
Absolute muscle power	−0.00 (−0.00, −0.00)	0.000	−0.00 (−0.00, −0.00)	0.000	−0.00 (−0.00, −0.00)	0.000
Relative muscle power	−0.74 (−0.97, −0.52)	0.000	−0.60 (−0.83, −0.37)	0.000	−0.37 (−0.52, −0.22)	0.000
Allometric muscle power	−0.01 (−0.02, −0.00)	0.000	−0.02 (−0.02, −0.01)	0.000	−0.01 (−0.01, −0.00)	0.000
6 min walking test						
Countermovement jump	2.46 (0.80, 4.12)	0.000	1.55 (−0.14, 3.24)	0.073	3.77 (1.17, 6.37)	0.005
5-time sit-to-stand	−16.5 (−22.2, −10.7)	0.000	−15.2 (−20.9, −9.6)	0.000	−17.1 (−24.2, −10.0)	0.000
Absolute muscle power	0.32 (0.14, 0.51)	0.000	0.41 (0.21, 0.61)	0.000	0.44 (0.20, 0.67)	0.000
Relative muscle power	42.3 (25.8, 58.8)	0.000	36.0 (19.2, 52.8)	0.000	37.6 (18.1, 57.0)	0.000
Allometric muscle power	0.77 (0.26, 1.29)	0.003	1.15 (0.57, 1.72)	0.000	1.16 (0.50, 1.83)	0.001
Walking speed at usual pace						
Countermovement jump	−0.00 (−0.03, 0.02)	0.000	−0.00 (−0.03, 0.02)	0.843	0.00 (−0.06, 0.06)	0.986
5-time sit-to-stand	−0.07 (−0.18, 0.03)	0.166	−0.07 (−0.18, 0.03)	0.169	−0.08 (−0.25, 0.08)	0.320
Absolute muscle power	−0.00 (−0.03, 0.04)	0.825	−0.01 (−0.00, 0.00)	0.708	0.01 (−0.00, 0.00)	0.737
Relative muscle power	0.11 (−0.19, 0.42)	0.473	0.12 (−0.19, 0.44)	0.428	0.16 (−0.29, 0.61)	0.486
Allometric muscle power	0.00 (−0.00, 0.01)	0.549	0.00 (−0.00, 0.01)	0.440	0.00 (−0.01, 0.02)	0.533
Walking speed at fast pace						
Countermovement jump	0.01 (−0.04, 0.07)	0.004	0.00 (−0.05, 0.06)	0.787	0.02 (−0.07, 0.01)	0.614
5-time sit-to-stand	−0.09 (−0.30, 0.10)	0.316	−0.09 (−0.30, 0.11)	0.386	−0.08 (−0.36, 0.20)	0.572
Absolute muscle power	−0.00 (−0.00, 0.00)	0.788	0.00 (−0.00, 0.00)	0.923	0.00 (−0.00, 0.00)	0.980
Relative muscle power	0.14 (−0.44, 0.72)	0.640	0.15 (−0.45, 0.76)	0.610	0.13 (−0.62, 0.90)	0.727
Allometric muscle power	0.00 (−0.01, 0.01)	0.980	0.00 (−0.01, 0.02)	0.683	0.00 (−0.02, 0.02)	0.821

^1^ Model 1: adjusted for age, body mass index, and sex; ^2^ Model 2: adjusted for Model 1 plus hypertension, diabetes mellitus, joint pain, osteoarthritis, cardiovascular disease. CI, confidence interval.

## Data Availability

Data are available from the corresponding author upon reasonable request.

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
