# Peer review of "Agreement and Associations between Countermovement Jump, 5-Time Sit-To-Stand, Lower-Limb Muscle Power Equations, and Physical Performance Tests in Community-Dwelling Older Adults"

_jcm, 2024, doi:10.3390/jcm13123380_

Round 1

Reviewer 1 Report

Comments and Suggestions for Authors

Thank you for the opportunity to review this scientific paper with the title: Agreement Between

Countermovement Jump, 5-Time Sit-To-Stand, and Lower-Limb Muscle Power Equations in

Community-Dwelling Older Adults. The paper is well structured from a methodological point of

view, with a well-defined purpose and a proper analysis, but there would be a few moments to

examine:

- The title does not fully convey the idea, the purpose of the paper

- Less or no use of abbreviations in the abstract

- The introduction gives a general overview of the importance of muscle strength in older

people but does not develop and exemplify in more detail information from the literature and

its importance to the study.

- The selection criteria for participants are too broad, which could influence the results.

- I would recommend a more detailed justification of the choice of statistical methods.

- Weak correlations were identified between the tests used, but the clinical implications of

these findings and their clinical importance or future research directions were not explored in

depth.

- The significant associations found between measures of muscle strength and physical

performance tests, such as the IHG, TUG and 6MWT, are interesting but only superficially

discussed.

- In section: Discussion literature is often cited, but in most cases it does not fit the topic and

results presented.

- The conclusion is too brief and does not report the main idea of the study.

Author Response

Reviewer 1

Thank you for the opportunity to review this scientific paper with the title: Agreement Between Countermovement Jump, 5-Time Sit-To-Stand, and Lower-Limb Muscle Power Equations in Community-Dwelling Older Adults. The paper is well structured from a methodological point of view, with a well-defined purpose and a proper analysis, but there would be a few moments to examine:

- The title does not fully convey the idea, the purpose of the paper

Answer: Thank you for highlighting this important aspect. The title has been revised to more accurately reflect the objective of our investigation.

- Less or no use of abbreviations in the abstract

Answer: Thank you for your comment. In the original submission, we used abbreviations to stay within the word count limit for the abstract. However, we agree with your point and have eliminated most abbreviations as recommended.

- The introduction gives a general overview of the importance of muscle strength in older

people but does not develop and exemplify in more detail information from the literature and its importance to the study.

Answer: We would like to thank the Reviewer for the comment. The introduction section has been revised. We have explained that muscle power is associated with many health parameters and might predict negative events in advanced age. As such, its active monitoring in older adults has been highly recommended. We have also explained that, although some assessment tools are available, they do not seem to be comparable proxies of muscle power. Specifically, previous studies noted that 5STS muscle power equations were poorly associated with other measures of muscle power. Furthermore, muscle power must be associated with physical performance, given that it is acknowledged as a candidate proxy of overall physical performance. However, some studies have reported no associations between 5STS muscle power equations and physical performance.  All these considerations have now been included in the introduction section which has allowed us to better frame the rationale of our study.

- The selection criteria for participants are too broad, which could influence the results.

Answer: Thank you for this comment. We agree that the heterogeneity of participants might influence the results. However, because the ultimate aim of our study was to provide practical information on the relationship between muscle power measures and physical performance in older adults, we decided not to apply selection criteria in order to recruit a “real world” sample of old individuals. We believe that our choice not to apply stringent inclusion/exclusion criteria is consistent with the objective of providing researchers and healthcare professionals with pragmatic information on the caveats of muscle power assessment in older adults.

- I would recommend a more detailed justification of the choice of statistical methods.

Answer: Thank you for this request. As recommended, justifications of the use of individual tests have been included in the statistical analysis section.

- Weak correlations were identified between the tests used, but the clinical implications of these findings and their clinical importance or future research directions were not explored in depth.

Answer: Thank you for highlighting this important point. Following your remark, we have now described the potential implications of our findings in lines 315-323.

- The significant associations found between measures of muscle strength and physical performance tests, such as the IHG, TUG and 6MWT, are interesting but only superficially discussed.

Answer: Thank you for this pertinent comment. We have carefully revised the discussion section and provided a more detailed discussion on the associations between physical performance tests and muscle power.

- In section: Discussion literature is often cited, but in most cases it does not fit the topic and results presented.

Answer: Thank you for this remark. As mentioned in the previous response, the discussion section has been thoroughly revised and relevant literature has been cited.

- The conclusion is too brief and does not report the main idea of the study.

Answer: We agree with this remark and have expanded the conclusion section as recommended.

Reviewer 2 Report

Comments and Suggestions for Authors

This study approaches the evaluation of distinct kinesthetic parameters in elders by applying different tests involving limb strength, balance and gait.

Even though the study has large number of participants, and performs a good number of tests, the major flow is the lack of a clinical goal.

The performance of all those tests needs to be connected with a proper clinical aim that gives sense to the clinical usefulness for their application. I really don´t understand why it would be meaningful to analyze the association of the tests results with each other, and not with clinical parameters such as, co-morbidities, BMI, age, or even socioeconomic parameters (educational level, place of residence, etc.  The performance of this tests should be useful either to detect or further study some clinical aspects of the individuals. If the only aim of the study is the validation of the tests and the association to each other, then another journal should be chosen and maybe focus on how the tests are performed and which one is faster or gives better results. In terms of clinical medicine, this study lacks of clinical correlation which is necessary not only to publish here but to give a meaningful contribution on the field.

The paper should be improved completely in order to insert the clinical parameter that is going to be approached through the tests.

Author Response

Reviewer 2

This study approaches the evaluation of distinct kinesthetic parameters in elders by applying different tests involving limb strength, balance and gait. Even though the study has large number of participants, and performs a good number of tests, the major flow is the lack of a clinical goal. The performance of all those tests needs to be connected with a proper clinical aim that gives sense to the clinical usefulness for their application. I really don´t understand why it would be meaningful to analyze the association of the tests results with each other, and not with clinical parameters such as, co-morbidities, BMI, age, or even socioeconomic parameters (educational level, place of residence, etc.  The performance of this tests should be useful either to detect or further study some clinical aspects of the individuals. If the only aim of the study is the validation of the tests and the association to each other, then another journal should be chosen and maybe focus on how the tests are performed and which one is faster or gives better results. In terms of clinical medicine, this study lacks of clinical correlation which is necessary not only to publish here but to give a meaningful contribution on the field. The paper should be improved completely in order to insert the clinical parameter that is going to be approached through the tests.

Answer: We would like to thank the Reviewer for raising these important points. We realize that in our original submission the clinical implications of our research were not sufficiently highlighted. Following your suggestions, we have carefully revised the introduction and discussion section to highlight the clinical relevance of muscle power and physical performance measures in older adults, the need for reliable tests to assess these domains, and the relationships among existing methods. In the revised introduction section, we have discussed in more detail the relevance of muscle power, muscle strength, and physical performance as predictors of poor outcomes in older adults. We have also indicated that these measures are widely used in geriatrics and gerontology to identify individuals at risk of negative events (e.g., falls, hospitalization), geriatric syndromes (e.g., frailty), and neuromuscular diseases (e.g., sarcopenia). The premises behind our study, as described in the introduction section, is that there is no agreement regarding the best approach to assess muscle power in clinical practice and research. Recently, muscle power equations were proposed, but the validity of these measures was only partially explored. The countermovement jump test, on the other hand, is a valid measure of muscle power, but might have limitations in clinical applicability. Based on these considerations, we have compared the two measures to examine whether they may be comparable proxies of muscle power. An important characteristic of muscle power, mentioned in the introduction section, is that muscle power may serve as an indicator of overall physical performance in older adults. This implies that a valid measure of muscle power should be associated with the results of as many physical performance tests as possible. That is the reason why we conducted a large analysis to explore associations between muscle power measures and physical performance tests. Hence, in our view, the study provides important information to clinicians and researchers by indicating that existing tools for estimating muscle power probably capture distinct facets of this physical capacity. Because regular monitoring of muscle power is essential to timely identify older adults at risk of negative events, we are confident that health professionals responsible for the care of older adults will find the results of our study relevant to clinical practice. In addition, our findings are expected to prompt researchers in the field to conduct further studies to better establish the validity of available instruments and explore potential adjustments that could enhance their accuracy. Following your comments, all these considerations have been included in the revision to better frame the clinical and research relevance of our study.

Reviewer 3 Report

Comments and Suggestions for Authors

This study, which examines the associations between the 5-Time Sit-To-Stand (STS) test, the countermovement jump (CMJ) test, and lower limb muscle strength, is valuable due to its relatively large sample size. However, insufficient analysis and presentation methods

-I could not find Figure 1 in the manuscript.

-In multiple regression analysis, it is more common to report standardized beta coefficients rather than unstandardized beta coefficients. This is because standardized beta coefficients allow for the comparison of the relative influence of different variables. Additionally, please include p-values to indicate whether the relationships between the variables are significant.

-The participants are predominantly female. Considering that men and women can be viewed as different entities in terms of physical function, it is necessary to include at least stratified analyses by gender.

-How was the complication information for each participant collected? What are the definitions of these complications?

Comments on the Quality of English Language

None.

Author Response

Reviewer 3

This study, which examines the associations between the 5-Time Sit-To-Stand (STS) test, the countermovement jump (CMJ) test, and lower limb muscle strength, is valuable due to its relatively large sample size. However, insufficient analysis and presentation methods

I could not find Figure 1 in the manuscript.

Answer: We apologize for the oversight. We have now included Figure 1 directly in the manuscript.

In multiple regression analysis, it is more common to report standardized beta coefficients rather than unstandardized beta coefficients. This is because standardized beta coefficients allow for the comparison of the relative influence of different variables. Additionally, please include p-values to indicate whether the relationships between the variables are significant.

Answer: Thank you for your comment. We respectfully disagree with your remark that standardized beta is preferentially used in comparison to unstandardized β. In the present study, in particular, we believe that it is more important to understand the effects of 1 unit change on variables X, Y, and Z than the impact of each variable. As requested, P values have been included in Table 2.

The participants are predominantly female. Considering that men and women can be viewed as different entities in terms of physical function, it is necessary to include at least stratified analyses by gender.

Answer: We agree with your comment. Model 1 of the regression analysis has been conducted according to sex (men/women) as a covariate.

How was the complication information for each participant collected? What are the definitions of these complications?

Answer: Thank you for this comment. The present study followed a cross-sectional design. Therefore, we did not examine the incidence of complications. If the Reviewer is referring to the methodology used to collect information on the presence of specific diseases, a description is available in lines 114-116.

i.e., Information on medical conditions was gathered through self-reporting and a careful examination of the community senior center’s medical charts.

Round 2

Reviewer 1 Report

Comments and Suggestions for Authors

accept as it

Reviewer 2 Report

Comments and Suggestions for Authors

I am satisfied with the response. Thank you. 

Reviewer 3 Report

Comments and Suggestions for Authors

Indeed, the authors have sincerely addressed my comments. I have no further requests for revisions. However, I emphasize that this paper does not meet the standards required for publication in JCM.